# A generative foundation model for antibody sequence understanding

**Justin Barton   Aretas Gaspariunas   David A. Yadin   Jorge Dias   Francesca L. Nice   Danielle H. Minns**
**Olivia Snudden   Chelsea Povall   Sara Valle Tomas   Harry Dobson   James H. R. Farmery**
**Jinwoo Leem   Jacob D. Galson** [1]

## Abstract

Here we introduce FAbCon, a generative antibody-specific language model comprising 2.4 billion parameters. A commonly accepted wisdom in developing large language models is that increasing model scale will translate to higher performance on downstream tasks. Starting from a 144-million parameter setup, we show that progressively larger models achieve greater accuracy in predicting antigen binding and can also be used to design new antibodies with good predicted developability potential. FAbCon is available on `huggingface.co/alchemab`.

## 1. Introduction

The human antibody repertoire represents an atypical and exceptionally diverse subset of protein sequences. Humans can theoretically express over $10^{15}$ unique antibodies to cover the wide breadth of antigens they will encounter in their lifetime (Briney et al., 2019). Deciphering the information encoded in the human antibody repertoire offers promise for better understanding of disease, improved diagnostics, and enabling therapeutic discovery (Bashford-Rogers et al., 2019; Galson et al., 2020; Greiff et al., 2020; Leem et al., 2022). Large language models (LLMs) have emerged as a tool to process and learn patterns from huge datasets of antibody repertoires, leading to breakthroughs in antibody structure prediction, antibody function prediction, and generative antibody design (Barton et al., 2024; Bachas et al., 2022; Burbach & Briney, 2024; Chen et al., 2024; Kenlay et al., 2024; Nijkamp et al., 2023; Ruffolo et al., 2023; Shuai et al., 2023).

The typical paradigm for antibody sequence analysis via LLMs involves two steps. First, the LLM is pre-trained on a large dataset of antibody sequences by masked language modelling or causal language modelling (CLM). After pre-training, the model is specialised for individual tasks by fine-tuning, or the pre-trained model is immediately applied for predictions without further tuning – also known as zero-shot learning. Currently, there is no consensus on the optimal pre-training objectives, datasets, and design decisions for developing and evaluating antibody-specific LLMs (Leem et al., 2022; Nijkamp et al., 2023; Bachas et al., 2022; Burbach & Briney, 2024; Ruffolo et al., 2021; Prihoda et al., 2022; Shanehsazzadeh et al., 2024; Olsen et al., 2022; Hie et al., 2024).

An additional bottleneck for progress has been the scale of compute. In natural language processing applications, LLMs leverage billions of parameters to maximise performance (Hoffmann et al., 2022). However, training antibody-specific LLMs at such ambitious levels has been challenging due to the paucity of hardware. In addition, antibody-specific LLMs have seen performance plateau, or even degrade, beyond 110 million parameters (Bachas et al., 2022; Nijkamp et al., 2023).

We introduce FAbCon, an antibody-specific LLM comprising 2.4 billion parameters. Based on the Falcon LLM from natural language processing (Almazrouei et al., 2023), and akin to other generative protein LLMs such as ProGen2 and ProTGPT2 (Nijkamp et al., 2023; Ferruz et al., 2022), FAbCon is pre-trained using CLM on unpaired and paired antibody sequences (Figure 1A; see Methods). After pre-training, FAbCon develops a rich representation of antibody sequences that can be leveraged for an extensive range of tasks; here, we focus on antigen binding prediction and antibody design. FAbCon-small (144 million parameters), FAbCon-medium (297 million parameters) and FAbCon-large (2.4 billion parameters) are available in our Hugging Face repository. Each variant can accept heavy-chain only, light-chain only, or paired chain inputs.

## 2. Methods

### 2.1. Datasets

FAbCon was pre-trained using CLM on an antibody-specific corpus, comprising both paired (i.e. heavy and light chains

---

[1] Alchemab Therapeutics Ltd, 1 Lion Works, Station Road East, Whittlesford, CB22 4WL. Correspondence to: Jacob D. Galson <jake@alchemab.com>.

*Accepted at the 1st Machine Learning for Life and Material Sciences Workshop at ICML 2024.* Copyright 2024 by the author(s).

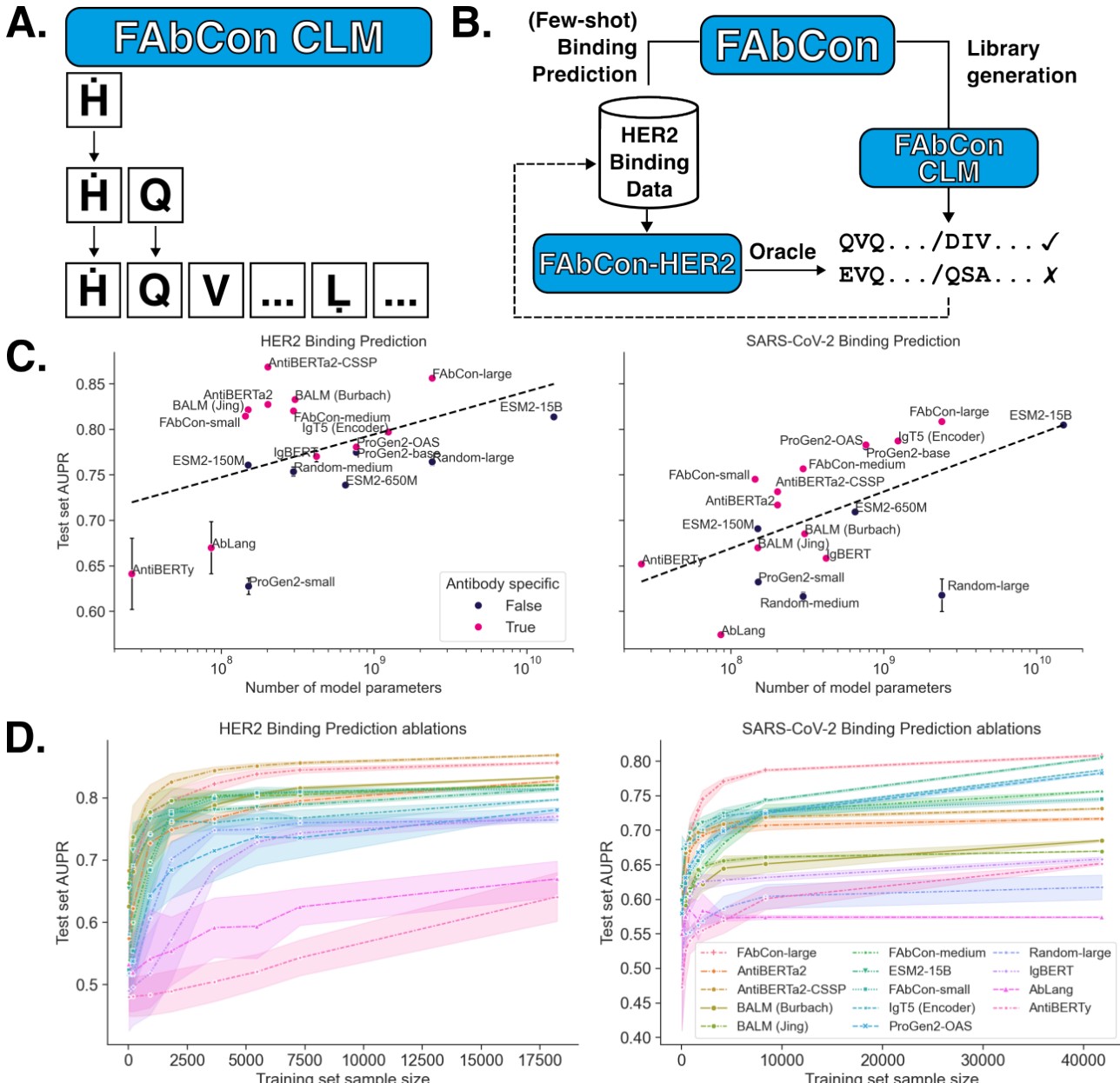

*Figure 1.* FAbCon is a generative, antibody-specific LLM, capable of sequence understanding. A. FAbCon is pre-trained by causal language modelling (CLM). FAbCon learns to predict the next amino acid using only the preceding N-terminal residues. By default, FAbCon generates paired antibody sequences. B. FAbCon is a foundation model that can be fine-tuned for library generation or few-shot antibody-antigen binding prediction. After fine-tuning for antibody-antigen binding, the model can act as an 'oracle' to screen libraries generated in vivo, in vitro, or in silico by a generative FAbCon model. Screened variants can then supplement binding datasets to 'close the loop' for antibody engineering. C. Area under the precision recall (AUPR) score of LLMs on HER2 and SARS-CoV-2 binding prediction versus the log of the number of model parameters. For IgT5, only the encoder module is publicly available, and we plot the number of IgT5's encoder parameters here. Antibody-specific LLMs are marked in pink. Error bars are standard deviations across five random seeds. Dotted line represents a line of best fit between the log of the number of parameters versus AUPR. D. HER2 and SARS-CoV-2 binding prediction AUPR as a function of training set sample size. Shaded areas represent standard deviations from five different seeds for fine-tuning.

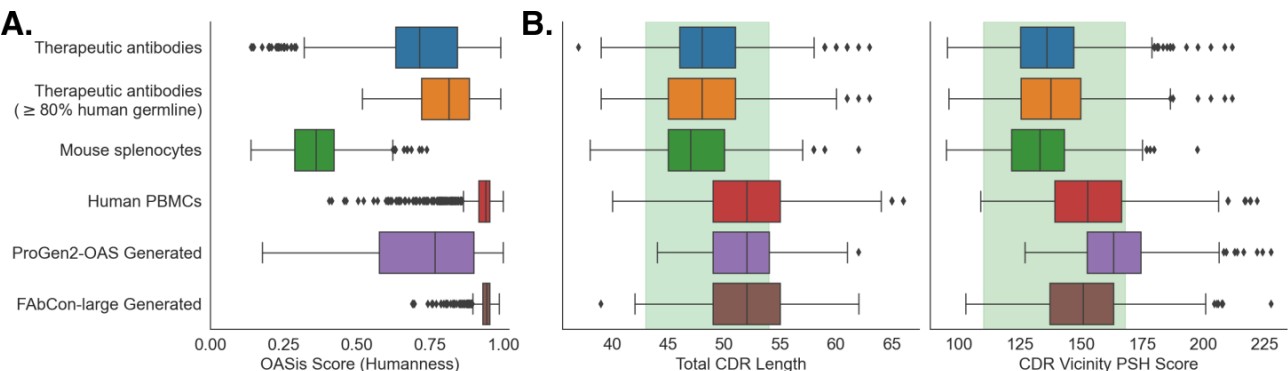

*Figure 2.* Computational developability assessment of generated antibody sequences. Boxes indicate the median, first and third quartiles; whiskers are 1.5x the inter-quartile range. A. OASis scores approximate antibody humanness. B. Distribution of two TAP metrics; shaded green areas indicate acceptable regions. Colour scheme and ordering are identical to A.

in one single sequence) and unpaired (i.e. heavy or light chain only) antibodies. Further details are described in Section A.1.

Three antibody-antigen binding datasets were collated for fine-tuning LLMs on antigen binding prediction: antibodies binding HER2, antibodies binding SARS-CoV-2 peptide, and antibodies binding IL-6 (Mason et al., 2021; Engelhart et al., 2022; Tsuruta et al., 2023). For each dataset, we split antibodies into disjoint train and test sets, with 80% of the data allocated for training, 10% for validation, and 10% for testing. Additional details of each antibody-antigen binding set are described in Section A.2.

Sequence generation was benchmarked against paired antibody sequences from three separate datasets; this is further described in Section A.3.

### 2.2. FAbCon pre-training and fine-tuning

FAbCon is a decoder-only transformer model based on the Falcon LLM (Almazrouei et al., 2023). FAbCon was pre-trained using the standard CLM objective. For all three antigens (HER2, SARS-CoV-2, IL-6), antibody-antigen binding prediction was framed as a binary classification task. Full details on pre-training and fine-tuning are described in Section A.4.

### 2.3. Sequence generation and computational developability estimation

One thousand unique paired antibody sequences were sampled using FAbCon-large or ProGen2-OAS. For both models, sequences were generated using nucleus sampling, with a temperature of 1.0 and a top-p value of 0.95. Developability profiles were computationally measured using OASis and TAP (Prihoda et al., 2022; Raybould et al., 2019a). Further

details on generation and the computational developability estimators are described in Sections A.7, A.8.

## 3. Results

In developing antibody therapeutics, the primary feature of interest is an antibody's binding to its cognate antigen. A model that can reliably predict antigen binding can be used as an 'oracle' to identify binders and enrich binding antibodies from B cell receptor (BCR) repertoire sequencing libraries, or from synthetic libraries. Following screening, data can then be fed back to 'close the loop' and enhance the oracle (Figure 1B). We fine-tuned our three FAbCon variants for binding prediction on three antigens: HER2, SARS-CoV-2 spike peptide, and IL-6 (Mason et al., 2021; Engelhart et al., 2022; Tsuruta et al., 2023). In addition, we also fine-tuned five general protein LLMs and nine antibody-specific LLMs (Figure 1C, Tables B1, B2) (Barton et al., 2024; Lin et al., 2023; Ruffolo et al., 2021; Shuai et al., 2023; Jing et al., 2023; Burbach & Briney, 2024; Kenlay et al., 2024; Nijkamp et al., 2023; Olsen et al., 2022). For each fine-tuning run, we freeze the encoder or decoder blocks and only allow weight updates in the classification head to better distinguish the quality of the underlying sequence representations of each LLM.

Overall, FAbCon-large has the strongest performance, as measured by the area under the precision recall curve (AUPR) score and area under the receiver operating curve scores (Figure 1C, Tables B1, B2). FAbCon-large has a mean AUPR of 0.883 across all three antigens; the closest competitor is ProGen2-OAS, with mean AUPR=0.847. We also randomly initialised and fine-tuned a FAbCon-large and a FAbCon-medium model without any pre-training, which we refer to as 'random-large' and 'random-medium'. For all antigens, random-large and random-medium signifi-

cantly underperformed pre-trained FAbCon models (mean AUPR=0.765, mean AUPR=0.719), reinforcing the value of pre-training.

Three themes are corroborated across our antigen binding benchmarks. First, model scale indeed translates to stronger performance. Within each of the ESM2, ProGen2, and FAbCon model families, model scaling yields better returns (Figure 1C, Tables B1, B2). Second, when normalising for model scale, antibody-specific LLMs outperform general protein LLMs. For example, ProGen2-OAS and ProGen2-base have identical numbers of parameters, yet ProGen2-OAS outperforms ProGen2-base. Finally, FAbCon models outperform larger models, indicating opportunities for greater accuracy at smaller scale.

We also measured the impact of the size of the training set by conducting ablation experiments, ranging from 0.1% to 100% of the HER2 and SARS-CoV-2 datasets (Figure 1D, see Methods). With only 5% of the HER2 training data (912 examples), FAbCon-large has AUPR=0.775±0.007, compared to AUPR=0.856±0.001 for the full dataset, and AUPR=0.607 for the random-large model. At 0.1% of the data (19 examples), no model achieves AUPR above 0.7. We investigate the feasibility of zero-shot prediction by using sequence perplexity to predict affinity; unlike previous studies, we report perplexity to be a poor discriminator for binders (Figure B1). Thus, LLMs can only become reliable antibody engineering oracles once a minimum volume of antigen binding data is available.

An exciting potential of generative models such as FAbCon is the ability to design antibody sequences. Owing to its decoder architecture, FAbCon can generate variable length sequences or infill variable length spans. Provided FAbCon can generate sequences with desirable properties (e.g. low immunogenicity), FAbCon can be used as a library generator and a computational panning tool (Figure 1B).

To test the validity of generative models for antibody design, we generated 1000 unique paired antibody sequences *de novo* using FAbCon-large. In addition, 1000 paired antibodies were synthesised via ProGen2-OAS using prompts (see Section A.7). We calculated the OASis score as a measure of humanness (Prihoda et al., 2022) for generated antibodies, 715 therapeutic antibodies, as well as 1000 BCRs from a mouse, and 1000 BCRs from a healthy human donor. For each set, we also used TAP (Raybould et al., 2019a) to estimate developability profiles.

The OASis scores of FAbCon-large generated antibodies resemble the distribution of OASis scores from a human BCR repertoire (Figure 2A). As expected, antibodies from ProGen2-OAS have lower OASis scores; this is likely due to ProGen2-OAS being pre-trained on BCR repertoires of multiple species. While therapeutic antibodies' OASis scores

are closer to those of ProGen2-OAS, it is worth noting that many antibodies in the clinic have murine origins and are humanised (e.g. pembrolizumab). Indeed, therapeutic antibodies with 80% sequence identity to human germline V genes (434 of 715 therapeutics) have higher OASis scores. In fact, FAbCon-large's sequence perplexities correlate strongly with OASis scores (Pearson r=-0.938, Figure B2), indicating FAbCon can be used as a humanness oracle zero-shot.

The TAP profiles of FAbCon reside in TAP's acceptable window (Figures 2B, B3). Generated antibodies from FAbCon-large feature longer complementarity-determining region (CDR) loops and more hydrophobic patches, which are known characteristics of human BCRs (Raybould et al., 2019a). Across the three CDR loops in the heavy chain, FAbCon-large's sequence length distributions most closely match human PBMCs (Figure B4).

## 4. Conclusion

Generative foundation models offer promise both as a tool to understand the information embedded within antibody sequences, and for designing new antibodies that can transform medicine. With increasing model scale and extensive evaluations, we demonstrate FAbCon's suitability to facilitate antibody engineering; we release all three of our FAbCon models under a modified Apache 2.0 license. We hope that FAbCon will inspire a more open sharing of data to continuously assess and improve LLMs for antibody sequence understanding and design. For future work, we envisage controlled generation to create better lead molecules.

## Acknowledgements

The authors would like to thank Bruno Trentini, Hassan Sirelkhatim, Christian Dallago, Maxine Kennedy, Lee Carter, and the wider NVIDIA team for their support. The authors would also like to thank Will Shaw and Camillo Anania from AWS for their support.

## Contributions

J.B. and J.L. conceived and designed the experiments. J.B. and J.L. performed the experiments. J.B., A.G. and J.L. processed the datasets for the study. J.H.R.F. and H.D. helped set up cloud and security infrastructure. D.A.Y., D.H.M., C.P., S.V.T. prepared the samples for single-cell sequencing of Alchemab's proprietary data. D.A.Y. and S.V.T. performed the single-cell sequencing. J.D., F.N., O.S., S.V.T. prepared and sequenced the samples for bulk sequencing of Alchemab's proprietary data. J.B., J.L., J.D.G. analysed the results. J.B., J.L., J.D.G. wrote the manuscript. All authors reviewed the manuscript.

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

# A. Methods

## A.1. Datasets

### A.1.1. PRE-TRAINING DATASETS

The Observed Antibody Space (OAS) database was downloaded on 23rd February 2023. Samples were pre-filtered in a similar procedure as previously discussed (Shanehsazzadeh et al., 2024). For example, BCRs derived from pre-B-cell samples were not used. However, we retained all sequences regardless of count.

In total, we used 1.47 billion unpaired sequences (i.e., heavy chain or light chain only). In addition, we supplemented this corpus with a set of proprietary BCR sequences from 376 individuals, bringing the total number of sequences to 1.54 billion. We then used Linclust (Steinegger & Söding, 2018) to cluster the dataset at 90% sequence identity across the VH or VL domains. After clustering, 777.8 million sequences were retained from OAS, while 43.4 million sequences were retained from proprietary data.

In addition to the unpaired data, we also use heavy-light chain paired sequences for pre-training. We first combined 1.5 million publicly available paired sequences (Jaffe et al., 2022), and an in-house dataset of 1.4 million paired sequences. Due to the paucity of paired data, we applied a 99% redundancy criterion for paired data. This led to a new total of 2.5 million sequences, of which 1.4 million are from the Jaffe et al. (2022) dataset, and 1.1 million are from our in-house data.

The final dataset comprising of 823.7 million sequences (821.2 million unpaired, 2.5 million paired) was split into 95% (777 million unpaired sequences and 2.4 million paired sequences) and 5% (44.3 million unpaired sequences and 0.1 million paired sequences) for training and evaluating CLM, respectively.

FAbCon-small and FAbCon-medium were pre-trained using a modified version of the corpus that is only comprised of publicly available data.

## A.2. Fine-tuning datasets

### A.2.1. HER2 BINDING DATA

We downloaded and filtered a dataset comprising 39,108 variants of trastuzumab that were screened for binding to the human epidermal growth factor receptor 2 (HER2) protein (Mason et al., 2021). Briefly, the experiment used hybridomas expressing either complementarity determining region (CDR) H3 or CDRL3 variants, and antigen binding was confirmed via cell sorting. Antibodies that do not bind the antigen, i.e. negatives, were randomly under-sampled. The remaining antibodies were randomly split into disjoint train, test, and validation sets as done previously (Mason et al., 2021); 18223, 2278, and 2278 sequences were used for train, test, and validation, respectively.

### A.2.2. SARS-CoV-2 BINDING DATA

A dataset of 104,972 antibody interactions with a peptide in the HR2 region of the SARS-CoV-2 spike protein was used (Engelhart et al., 2022). First, mean predicted $\log K_D$ values were calculated across repeated observations for the same antibody sequence. Next, sequences with mean predicted $\log K_D < 3$ were designated binders, while sequences with mean predicted $\log K_D \geq 4$ were designated non-binders. Sequences with mean predicted $\log K_D \geq 3$ and $\log K_D < 4$ were designated ambiguous and discarded. The data were then randomly split into disjoint train, test, and validation sets of 41787, 5159, and 4644 sequences, respectively.

### A.2.3. IL-6 BINDING DATA

A dataset of 573,891 antigen-VHH pairs produced from an alpaca immunised with single-mutant variants of human interleukin-6 (IL-6) was used (Tsuruta et al., 2023). First, alpaca antibodies with higher similarity to human antibody germline sequences tended to bind IL-6. Thus, we removed any alpaca antibody with less than 75% sequence identity to a human V gene.

From the remaining 232,084 sequences, we then retained sequences that only bound one variant of the IL-6 molecule or did not bind any of the IL-6 variants, leading to 211,920 sequences. We then de-duplicated the antibody sequences; where an antibody was found to bind one variant of IL-6 but confirmed to not bind any others, we assumed it was a non-binder to reduce ambiguities. This led to a set of 14,467 sequences. Non-binder sequences were randomly under-sampled, leading to

a total of 1636 sequences.

The data were then randomly split into disjoint train, test, and validation sets of 1310, 163, and 163 sequences, respectively. We stratified the antibodies so that antibodies across the different sets have a minimum Levenshtein distance of 1 across the junction region.

### A.2.4. ABLATION EXPERIMENTS

For HER2 and SARS-CoV-2, each dataset was subsampled at different rates to test for the effects of training set size on model prediction performance. Subsamples ranging from 0.1% to 100% of the data were taken without replacement. At each sub-sampling rate, we trained models with five different random seeds to estimate variance.

### A.3. Control datasets for evaluating generative models

To assess the quality of generative models, we compiled three datasets as controls: a single-cell mouse BCR repertoire dataset from splenocytes (10x Genomics, 2020b), a single-cell human BCR repertoire dataset from PBMCs (10x Genomics, 2020a), and therapeutic antibodies from Thera-SAbDab (Raybould et al., 2019b).

The processed single-cell repertoire datasets were directly used from the 10x Genomics repository. A set of one thousand BCRs were randomly sampled from each set.

Thera-SAbDab was downloaded on March 27th, 2023. Only mono-specific antibodies and those that are mAbs or Fabs were used. Therapeutic antibodies were then de-duplicated by their V domain sequence. In total, 715 antibodies were used.

### A.4. FAbCon pre-training and fine-tuning

FAbCon is based on the Falcon LLM (Almazrouei et al., 2023). FAbCon-large was pre-trained using 48 NVIDIA A100s (80GB) from NVIDIA's DGX SuperCloud (Cambridge-1 environment), while FAbCon-small and FAbCon-medium were pre-trained using 48 NVIDIA A100's (80GB) from Amazon Web Services SageMaker. Full hyperparameters are described in Table B3.

Fine-tuning was done using NVIDIA H100s (80GB). For each LLM, including those without pre-training (e.g. random-large), the encoder or decoder weights were frozen. Predictions were then made on the corresponding test set of each antigen. Training was repeated using five different random seeds in order to estimate standard deviations.

Each FAbCon model has a maximum context window of 256, with a total of 26 tokens in its vocabulary: the 20 amino acids, `<|endoftext|>,<|unk|>,<|pad|>,<|mask|>`, along with $\dot{\text{H}}$ and $\underset{.}{\text{L}}$ to prefix heavy and light chains, respectively. During pre-training, both unpaired and paired antibody sequences can form one mini-batch.

FAbCon variants were pre-trained using the CLM objective, where the model is tasked with autoregressively predicting each residue by attending only to the previous residues in the sequence. The model parameters were updated via gradient-based optimisation in order to minimise the cross-entropy loss as given by

$$\mathcal{L}_{CE} = \frac{-1}{N} \sum_{t=1}^{T} \sum_{i=1}^{V} L_{t,i} \log P(r_t | r_1, r_2, \ldots r_{t-1})_i$$

where $N$ is the batch size, $T$ is the sequence length, $V$ is the vocabulary size, and $L_{t,i}$ is the label for the residue at position $t$, and the $i$th word in the vocabulary.

### A.5. Fine-tuning

Fine-tuning was done using NVIDIA H100s (80GB). Each LLM was fine-tuned on an antigen training set using a linear layer with a binary cross-entropy loss. Either the `[CLS]` token for encoder-based (e.g. ESM2) or the `<|endoftext|>` token for decoder-based (e.g. ProGen2) models were used as input for classification. For IgT5, we used mean pooling as previously described (Kenlay et al., 2024).

For each LLM, including those without pre-training (e.g. random-large), the encoder or decoder weights were frozen. Predictions were then made on the corresponding test set of each antigen. Training was repeated using five different random

seeds in order to estimate standard deviations. Hyperparameters for each antigen-specific task are listed in Table B4.

## A.6. Perplexity calculations

Perplexity scores were calculated as defined by

$$Perplexity(R) = \exp \frac{-1}{t} \sum_{i}^{t} \log p_\theta(r_i|r_{<i})$$

where $R$ is an antibody sequence, $t$ is the length of $R$, $r_1, \ldots r_t$ are the residues of $R$, and $\theta$ are the parameters of the model.

## A.7. Sequence generation

One thousand unique paired antibody sequences were sampled using FAbCon-large or ProGen2-OAS. For both models, sequences were generated using nucleus sampling (Holtzman et al., 2019), with a temperature 1.0, and a top-p value of 0.95.

For FAbCon-large, we used only the heavy chain prefix token (Ḣ) as the input. Sequences were sampled using a maximum sequence length of 256, which always synthesised paired antibody sequences. For ProGen2-OAS, heavy chains and light chains were independently generated using three different prompts. One thousand heavy chains were prompted with EVQ, while 500 light chains were prompted with QSA (lambda), and 500 light chains were prompted with DIQ (kappa), in-line with previous work (Shuai et al., 2023). Since ProGen2-OAS does not produce paired sequences, generated heavy and light chains were randomly paired.

## A.8. Computational developability estimation

Developability profiles of BCRs from repertoires, 715 therapeutic antibodies, and generated antibodies were computationally measured using OASis and TAP (Prihoda et al., 2022; Raybould et al., 2019a).

Briefly, OASis represents the fraction of overlapping 9-mers in an antibody sequence that is detected in at least 50% of human repertoires in the OAS database. Thus, an OASis score of 1.0 implies every possible 9-mer in the antibody sequence is found in at least 50% of humans. OASis scores were measured using BioPhi v1.0.9.

TAP encompasses five physicochemical and structural criteria: total CDR length, CDR vicinity patches of surface hydrophobicity (PSH), CDR vicinity patches of positive charge (PPC), CDR vicinity patches of negative charge (PNC), and variable fragment charge symmetry parameter (FvCSP). TAP profiles were measured using the online version of TAP. Each TAP metric has an acceptable range, which was last updated on July 1st, 2023 (Raybould et al., 2024). TAP failed to model 2 mouse antibodies, 479 antibodies from ProGen2-OAS, and 2 therapeutic antibodies (elipovimab and zinlirvimab).

# B. Results

*Table B1.* LLM performance on antigen-specific binding prediction measured by area under the precision-recall curve (AUPR) score. Each LLM is fine-tuned on the training set using a standard sequence classification head with the encoder or decoder weights frozen. Training is repeated for five different random seeds to estimate standard deviations. Antigen training dataset sizes are indicated with $N$.

| MODEL | TYPE | PARAMETERS | CORPUS SIZE | HER2 (N=18223) | SARS-CoV-2 (N=41787) | IL-6 (N=1310) |
|---|---|---|---|---|---|---|
| ESM2-150M | PROTEIN | 150M | 65M | $0.761 \pm 0.001$ | $0.691 \pm 0.002$ | $0.663 \pm 0.031$ |
| ESM2-650M | PROTEIN | 650M | 65M | $0.739 \pm 0.002$ | $0.709 \pm 0.001$ | $0.763 \pm 0.033$ |
| ESM2-15B | PROTEIN | 15B | 65M | $0.814 \pm 0.003$ | $0.805 \pm 0.001$ | $0.742 \pm 0.023$ |
| PROGEN2-SMALL | PROTEIN | 151M | 240M | $0.628 \pm 0.009$ | $0.632 \pm 0.000$ | $0.536 \pm 0.010$ |
| PROGEN2-BASE | PROTEIN | 765M | 240M | $0.775 \pm 0.002$ | $0.781 \pm 0.001$ | $0.966 \pm 0.003$ |
| RANDOM-MEDIUM (1024D) | RANDOM | 297M | 0 | $0.754 \pm 0.005$ | $0.616 \pm 0.004$ | $0.789 \pm 0.075$ |
| RANDOM-LARGE (2048D) | RANDOM | 2.4B | 0 | $0.764 \pm 0.004$ | $0.618 \pm 0.018$ | $0.912 \pm 0.055$ |
| ANTIBERTY | ANTIBODY | 26M | 558M | $0.645 \pm 0.044$ | $0.652 \pm 0.002$ | $0.881 \pm 0.124$ |
| ABLANG-HEAVY | ANTIBODY | 85M | 14.2M | $0.670 \pm 0.029$ | $0.574 \pm 0.001$ | $0.697 \pm 0.309$ |
| BALM (JING) | ANTIBODY | 148M | 336M | $0.822 \pm 0.001$ | $0.670 \pm 0.000$ | $0.631 \pm 0.026$ |
| BALM-PAIRED (BURBACH) | ANTIBODY | 304M | 1.3M | $0.833 \pm 0.001$ | $0.685 \pm 0.002$ | $0.564 \pm 0.012$ |
| IGBERT | ANTIBODY | 420M | 1.46B | $0.770 \pm 0.006$ | $0.658 \pm 0.003$ | $0.687 \pm 0.255$ |
| PROGEN2-OAS | ANTIBODY | 765M | 554M | $0.781 \pm 0.004$ | $0.783 \pm 0.001$ | $0.977 \pm 0.000$ |
| IGT5 (ENCODER) | ANTIBODY | 1.2B | 1.46B | $0.797 \pm 0.001$ | $0.787 \pm 0.000$ | $0.889 \pm 0.006$ |
| ANTIBERTA2 | ANTIBODY | 202M | 779.4M | $0.827 \pm 0.001$ | $0.717 \pm 0.002$ | $0.611 \pm 0.037$ |
| ANTIBERTA2-CSSP | ANTIBODY* | 202M | 779.4M | $0.869 \pm 0.001$ | $0.732 \pm 0.001$ | $0.985 \pm 0.001$ |
| FABCON-SMALL | ANTIBODY | 144M | 730.8M | $0.815 \pm 0.001$ | $0.745 \pm 0.002$ | $0.610 \pm 0.021$ |
| FABCON-MEDIUM | ANTIBODY | 297M | 730.8M | $0.820 \pm 0.002$ | $0.757 \pm 0.001$ | $0.627 \pm 0.077$ |
| FABCON-LARGE | ANTIBODY | 2.4B | 779.4M | $0.857 \pm 0.001$ | $0.809 \pm 0.002$ | $0.985 \pm 0.001$ |

*AntiBERTa2-CSSP is multimodal, using antibody structural data from the Protein Data Bank (Barton et al., 2024).

*Table B2.* LLM performance on antigen-specific binding prediction measured by area under the receiver operating curve (AUROC) score. LLM runs are described in Table B1.

| MODEL | TYPE | PARAMETERS | CORPUS SIZE | HER2 (N=18223) | SARS-CoV-2 (N=41787) | IL-6 (N=1310) |
|---|---|---|---|---|---|---|
| ESM2-150M | PROTEIN | 150M | 65M | $0.772 \pm 0.000$ | $0.725 \pm 0.001$ | $0.792 \pm 0.034$ |
| ESM2-650M | PROTEIN | 650M | 65M | $0.781 \pm 0.002$ | $0.736 \pm 0.001$ | $0.821 \pm 0.024$ |
| ESM2-15B | PROTEIN | 15B | 65M | $0.839 \pm 0.002$ | $0.821 \pm 0.001$ | $0.837 \pm 0.013$ |
| PROGEN2-SMALL | PROTEIN | 151M | 240M | $0.678 \pm 0.009$ | $0.663 \pm 0.000$ | $0.591 \pm 0.015$ |
| PROGEN2-BASE | PROTEIN | 765M | 240M | $0.798 \pm 0.001$ | $0.787 \pm 0.001$ | $0.946 \pm 0.005$ |
| RANDOM-MEDIUM (1024D) | RANDOM | 297M | 0 | $0.782 \pm 0.005$ | $0.659 \pm 0.001$ | $0.866 \pm 0.043$ |
| RANDOM-LARGE (2048D) | RANDOM | 2.4B | 0 | $0.794 \pm 0.004$ | $0.663 \pm 0.018$ | $0.926 \pm 0.022$ |
| ANTIBERTY | ANTIBODY | 26M | 558M | $0.666 \pm 0.044$ | $0.700 \pm 0.002$ | $0.905 \pm 0.075$ |
| ABLANG-HEAVY | ANTIBODY | 85M | 14.2M | $0.697 \pm 0.023$ | $0.615 \pm 0.001$ | $0.649 \pm 0.385$ |
| BALM (JING) | ANTIBODY | 148M | 336M | $0.848 \pm 0.001$ | $0.730 \pm 0.000$ | $0.745 \pm 0.019$ |
| BALM-PAIRED (BURBACH) | ANTIBODY | 304M | 1.3M | $0.842 \pm 0.001$ | $0.731 \pm 0.001$ | $0.664 \pm 0.017$ |
| IGBERT | ANTIBODY | 420M | 1.46B | $0.787 \pm 0.007$ | $0.699 \pm 0.001$ | $0.721 \pm 0.268$ |
| PROGEN2-OAS | ANTIBODY | 765M | 554M | $0.801 \pm 0.003$ | $0.788 \pm 0.001$ | $0.966 \pm 0.001$ |
| IGT5 (ENCODER) | ANTIBODY | 1.2B | 1.46B | $0.815 \pm 0.001$ | $0.792 \pm 0.000$ | $0.935 \pm 0.005$ |
| ANTIBERTA2 | ANTIBODY | 202M | 779.4M | $0.839 \pm 0.002$ | $0.738 \pm 0.002$ | $0.721 \pm 0.040$ |
| ANTIBERTA2-CSSP | ANTIBODY* | 202M | 779.4M | $0.875 \pm 0.001$ | $0.750 \pm 0.001$ | $0.976 \pm 0.001$ |
| FABCON-SMALL | ANTIBODY | 144M | 730.8M | $0.836 \pm 0.001$ | $0.767 \pm 0.001$ | $0.741 \pm 0.025$ |
| FABCON-MEDIUM | ANTIBODY | 297M | 730.8M | $0.831 \pm 0.001$ | $0.777 \pm 0.000$ | $0.742 \pm 0.082$ |
| FABCON-LARGE | ANTIBODY | 2.4B | 779.4M | $0.861 \pm 0.001$ | $0.815 \pm 0.001$ | $0.978 \pm 0.002$ |

*AntiBERTa2-CSSP is multimodal, using antibody structural data from the Protein Data Bank (Barton et al., 2024).

*Table B3.* Hyperparameter configuration for pre-training FAbCon LLMs.

|                            | FABCON-SMALL | FABCON-MEDIUM | FABCON-LARGE |
|----------------------------|:---:|:---:|:---:|
| NUMBER OF LAYERS           | 24 | 28 | 56 |
| NUMBER OF ATTENTION HEADS  | 12 | 16 | 32 |
| EMBEDDING DIMENSION        | 768 | 1024 | 2048 |
| FEED-FORWARD DIMENSION     | 3072 | 4096 | 8192 |
| TOTAL PARAMETERS           | 144M | 297M | 2.4B |
| PEAK LEARNING RATE         | | 2E-5 | |
| LEARNING RATE SCHEDULE     | | COSINE | |
| NUMBER OF STEPS            | | 200K | |
| OPTIMIZER                  | | FUSED ADAMW | |
| WEIGHT DECAY               | | 0.01 | |
| GRADIENT NORM CLIPPING     | | 1.0 | |
| MULTI-QUERY                | | TRUE | |
| PRE-TRAINING TASK          | | CLM | |

*Table B4.* Hyperparameter configuration for fine-tuning LLMs.

|                           | HER2 | SARS-COV-2 | IL-6 |
|---------------------------|:---:|:---:|:---:|
| BATCH SIZE                | 384 | 96 | 192 |
| EPOCHS                    | 12 | 15 | 14 |
| PEAK LEARNING RATE        | 1E-4 | 8E-5 | 1E-4 |
| NUMBER OF RANDOM SEEDS    | | 5 | |
| WARMUP RATIO              | | 0.15 | |
| LEARNING RATE SCHEDULE    | | LINEAR | |
| OPTIMIZER                 | | ADAMW | |

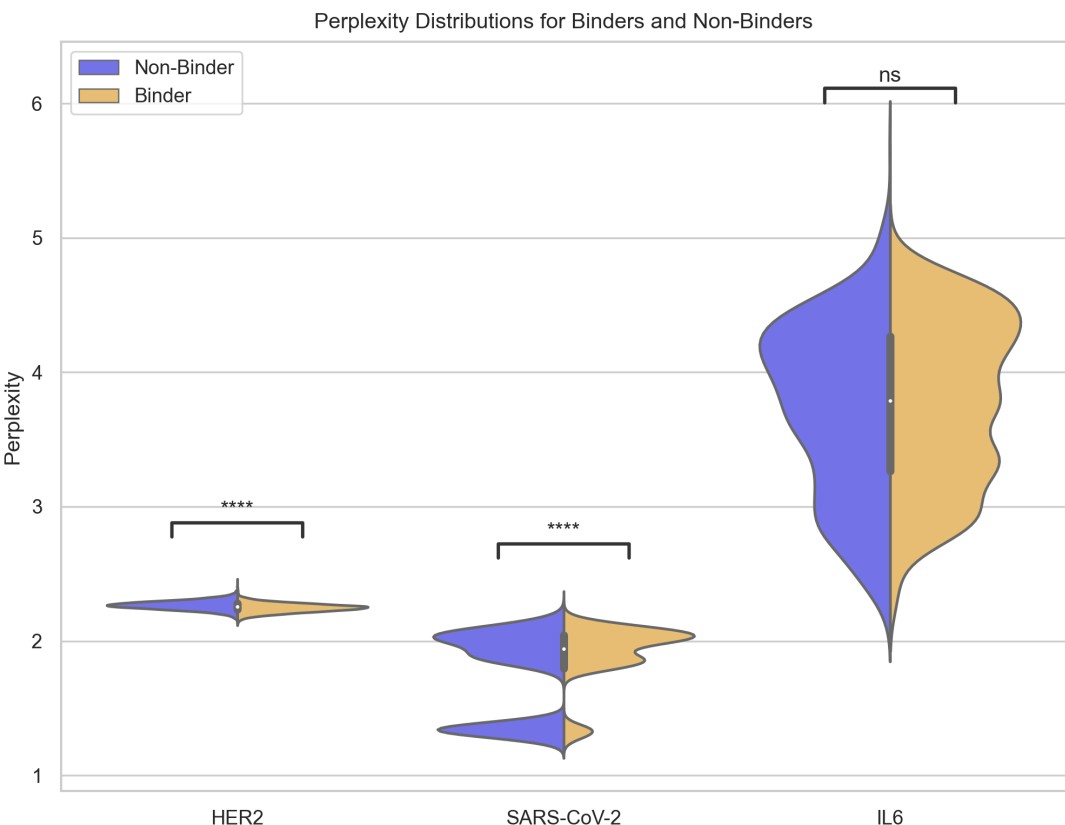

*Figure B1*. FAbCon-large perplexity scores for binding (orange) and non-binding (blue) antibodies against HER2, SARS-CoV-2, and IL-6. Significance levels above brackets indicate the results of a Mann-Whitney-Wilcoxon two-sided test with Bonferroni correction between the distribution of binders and non-binders for each antigen. Note that while there is a systematic difference in perplexity between binders and non-binders, the support of each distribution is nearly identical; classifying binding status by perplexity alone is intractable. ****: p $\leq$ 1e-4, ns: $0.05 < p \leq 1$.

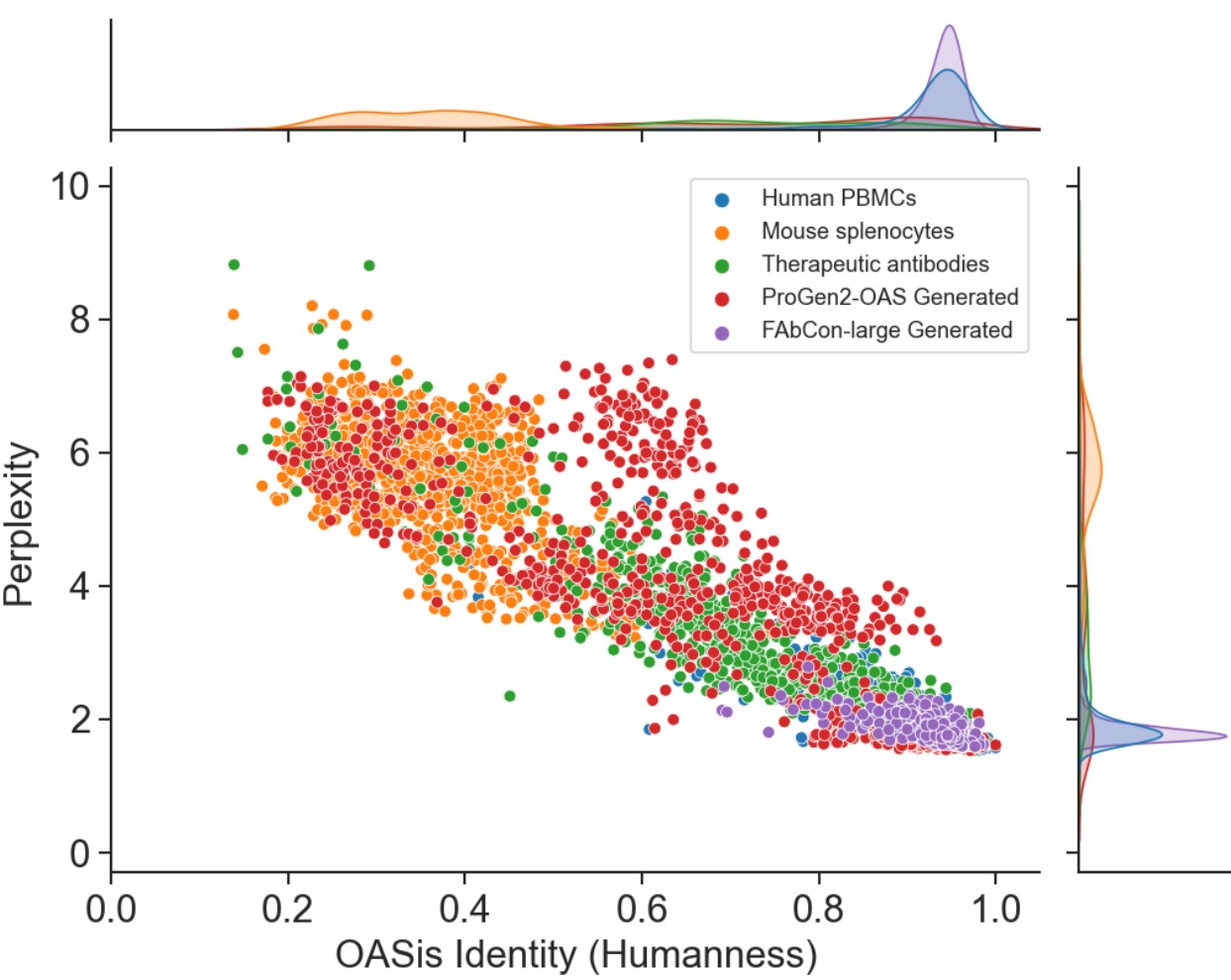

*Figure B2.* Perplexity is a zero-shot predictor of humanness. Perplexity is calculated for paired antibody sequences.

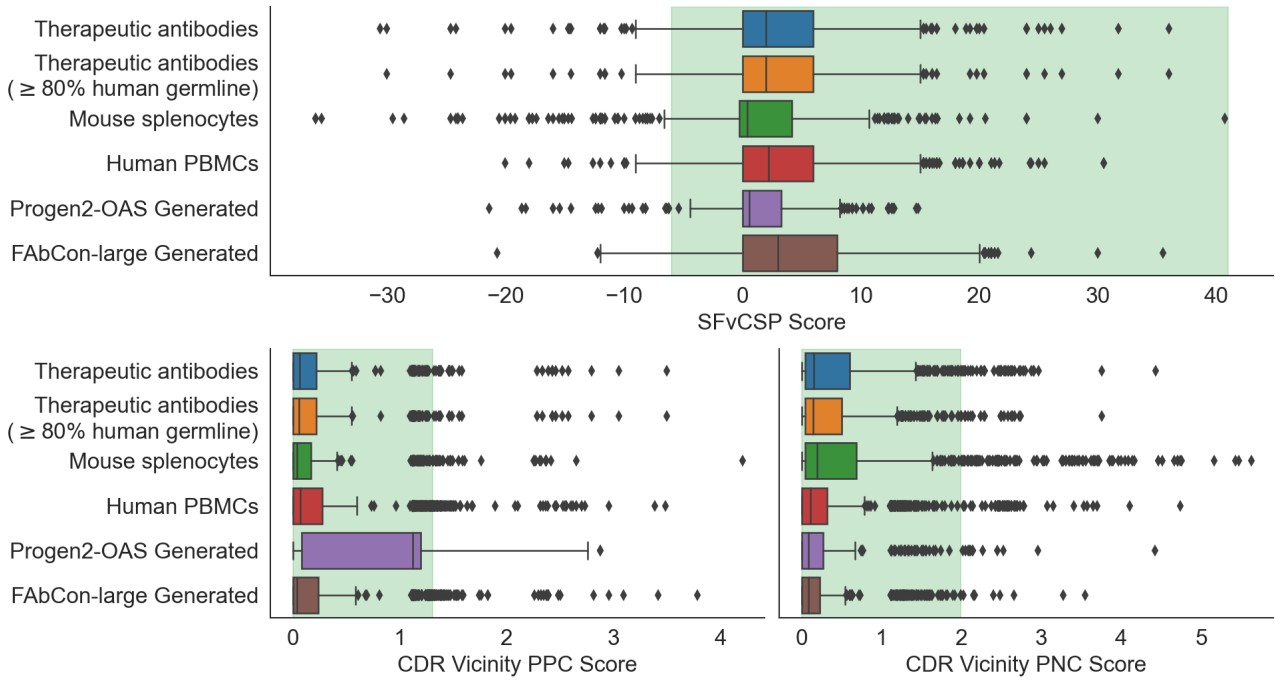

*Figure B3.* Distributions of TAP metrics. Shaded green areas indicate acceptable values for the different TAP metrics.

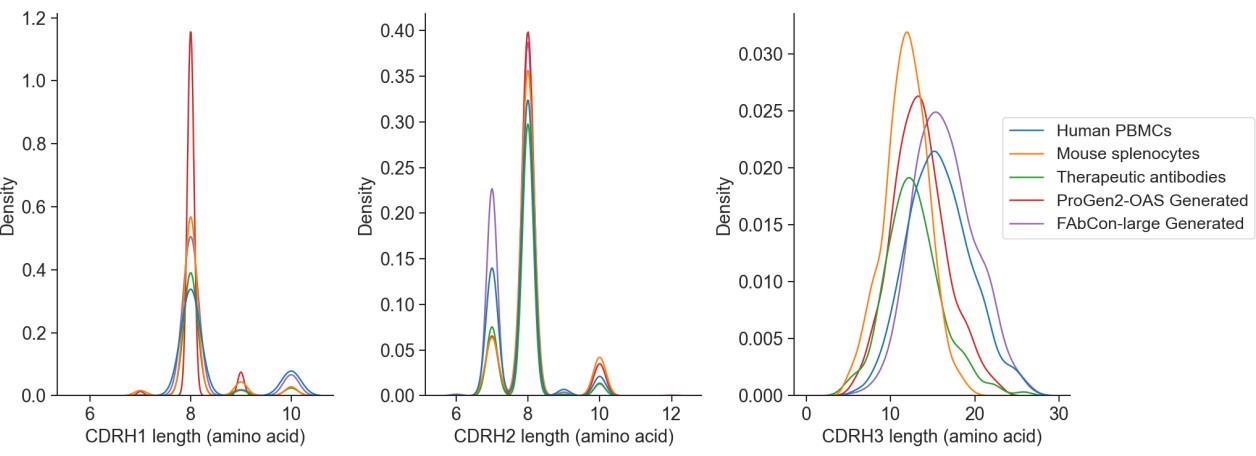

*Figure B4.* Distributions of heavy chain CDR amino acid sequence lengths from different antibody sources.