# OpenReview forum: "A generative foundation model for antibody sequence understanding"
_ICML.cc/2024/Workshop/ML4LMS — ML4LMS Poster_

### Official Review · Reviewer_ERcz · 2024-06-09
**Summary: The paper introduces FabCon LLMs for antibody sequences in 3 flavors: small, medium and large; pretrained on a mix of publicly available (OAS) and proprietary datasets. These LLMs when fine-tuned on antigen binding prediction and antibody design show promising results as compared to other approaches.**

**Rating:** 7
**Confidence:** 3

**Review:**

Overall it is a good paper, I’ve a few comments/questions:
- It seems that the small and medium versions are trained on smaller filtered dataset without proprietary data. I wonder if some of the results on finetuning tasks can be attributed to this aspect and if the authors explored this more. I believe the dataset aspect could also be highlighted in the results section “Within each of the ESM2, ProGen2, and FAbCon model families, model scaling yields better returns”, since model scaling + additional datasets seem to be the reason for the better results.
- I’m curious how quickly do the authors see plateauing of perplexity values vs training steps, during pretraining
- Figure B1 has a typo ****

---

### Official Review · Reviewer_Cfip · 2024-06-10
**Large decoder-only antibody language model**

**Rating:** 7
**Confidence:** 4

**Review:**

This work introduces FAbCon, an antibody-specific language model that is based on the decoder-only Falcon LLM. It is trained with a causal language modelling task, i.e. next token prediction conditioned on preceding amino acids in the sequence.
The authors train their model on a large corpus of public and proprietary data and show convincing benchmarks on two binding datasets that their approach can provide a competitive baseline.

Comments:
- It is not clear to me that a causal language modelling approach makes the most sense in the protein space, since in contrast to NLP there is no obvious reason why an amino acid appearing later in the sequence should depend on previous ones. It would be useful for the authors to expand on their choice of training objective, and the pros/cons this has compared to a more common MLM objective used by ESM.
- the author dedicate a full section to defining perplexity, but beyond a mention in the discussion this is not used anywhere in the main text, only in appendix figures, so it would make more sense to move this definition to appendix B

---

### Official Review · Reviewer_tUL3 · 2024-06-11
**FAbCon: A generative antibody language model with thorough benchmarking of affinity prediction finetuning.**

**Rating:** 7
**Confidence:** 4

**Review:**

The paper presents a generative (decoder-only) antibody language model, as well as a thorough benchmark of its performance for affinity prediction through fine-tuning. The evaluation of FAbCon's fine-tuned affinity prediction performance is extensive and novel, involving fine-tuning of several other LMs, and provides interesting insights into scaling laws, and zero-shot vs fine-tuning performance. However, the work could benefit from further evaluation of the sequences generated by FAbCon.

Suggestions:
1. Provide measures of the quality, diversity, and novelty of generated sequences and benchmark against other generative models.
2. Using standard characters to represent the heavy and light chain (e.g. '<heavy>') would improve the usability of the tool.
3. Benchmarking the performance of an LM that incorporates structural information (e.g. [SaProt](https://www.biorxiv.org/content/10.1101/2023.10.01.560349v5) for the fine-tuning task would be interesting.
4. Investigate using the affinity predictor to steer the generation of sequences (e.g. with [PPLMs](https://arxiv.org/abs/1912.02164)